# Serotonin Receptors as Therapeutic Targets for Autism Spectrum Disorder Treatment

**DOI:** 10.3390/ijms23126515

**Published:** 2022-06-10

**Authors:** Ansoo Lee, Hyunah Choo, Byungsun Jeon

**Affiliations:** 1Brain Science Institute, Korea Institute of Science and Technology, Seoul 02792, Korea; alee@kist.re.kr; 2Division of Bio-Medical Science & Technology, KIST School, University of Science and Technology, Seoul 02792, Korea

**Keywords:** serotonin receptors, autism spectrum disorders, therapeutic targets, modulators

## Abstract

Autism spectrum disorder (ASD) is a group of neurodevelopmental disorders characterized by repetitive and stereotyped behaviors as well as difficulties with social interaction and communication. According to reports for prevalence rates of ASD, approximately 1~2% of children worldwide have been diagnosed with ASD. Although there are a couple of FDA (Food and Drug Administration)—approved drugs for ASD treatment such as aripiprazole and risperidone, they are efficient for alleviating aggression, hyperactivity, and self-injury but not the core symptoms. Serotonin (5-hydroxytryptamine, 5-HT) as a neurotransmitter plays a crucial role in the early neurodevelopmental stage. In particular, 5-HT has been known to regulate a variety of neurobiological processes including neurite outgrowth, dendritic spine morphology, shaping neuronal circuits, synaptic transmission, and synaptic plasticity. Given the roles of serotonergic systems, the 5-HT receptors (5-HTRs) become emerging as potential therapeutic targets in the ASD. In this review, we will focus on the recent development of small molecule modulators of 5-HTRs as therapeutic targets for the ASD treatment.

## 1. Introduction

Autism spectrum disorder (ASD) is a complex neurodevelopmental disorder with heterogeneous etiology. Core symptoms of ASD include impaired social interactions, stereotyped repetitive behaviors, and restricted interests [1]. ASD is considered to be caused by genetic and environmental factors including maternal/paternal age, fetal environment, nutrition, and toxic exposures [2,3]. Since epidemiological investigation of ASD started in the mid-1960s [4,5], many countries have reported prevalence rates of ASD annually. The prevalence rates of ASD are notably becoming higher each year with present rates of approximately 1~2% worldwide, although there are a variety of reasons for this significant increase in the prevalence such as improved ASD diagnostic techniques and the rising awareness [6,7,8,9]. Given the growing concerns, development of treatments for ASD thus becomes a priority in recent public health issues. There are only a couple of FDA-approved drugs for ASD such as aripiprazole and risperidone [10,11,12,13,14]. They are certainly effective for alleviating the symptoms including aggression, hyperactivity and self-injury but not the ASD core symptoms as mentioned above. Effective therapeutics for the core symptoms thus remain underdeveloped to date due to lack of understanding etiology and pathological mechanisms for ASD [15].

Over the past decades, human clinical studies have been accumulated and several animal models for treatment of core symptoms in ASD have been developed [16,17]. Based on the development of these techniques, several potential mechanisms such as synaptic dysfunction (including excitatory/inhibitory imbalance in brain neural circuits and dendritic spine abnormalities) [18,19,20], neuroinflammation [21,22], and central neurotransmission systems [23,24,25] have been identified as therapeutic targets. Selective modulators of mGlu, NMDA, GABA receptors, and GSK-3β signaling pathway are known to regulate excitatory/inhibitory ratio, resulting in enhancement of neuroplasticity [18,19,20]. mTOR, ROCK kinases, and IGF1 system have been widely investigated to correct the dendritic spine abnormalities, while Toll-like receptors, involved in neuroinflammation, have been studied to modulate synaptic functions [21,22]. Serotonin as a key neurotransmitter plays a crucial role in the early neurodevelopmental stage. A variety of studies proposed that disturbances in serotonergic system are described in several neurodevelopmental disorders including ASD [25].

In particular, hyperserotonemia during pregnancy has been suspected as a potential environmental factor in the prevalence of ASD [26,27]. Selective serotonin reuptake inhibitors (SSRIs) that have been prescribed for the alleviation of depression antagonize serotonin transporter (SERT) and lead the increase of serotonin level in brain. Due to the potential connectivity, many studies have been reported the effect of SSRIs on ASD [15,28,29]. In 2011, Croen and co-workers investigated the effects of prenatal exposure to SSRIs on ASD risk. According to their population-based study (298 case children with ASD and 1507 control children), they found a two-fold increased risk of ASD with SSRIs exposure during prenatal period [30]. While more studies support the association, the opposite results have also been reported. Hviid and colleagues conducted a large number of cohort study between the use of SSRIs during pregnancy and ASD diagnosis in the offspring in Danish population [31]. They could not claim a significant association between the SSRIs exposure during pregnancy and ASD in the offspring. Payet and colleagues studied the period effect of SSRI exposure with young male BALB/c mice, a phenotype that is relevant to ASD [32]. The authors investigated the effects of chronic and acute administration of fluoxetine, showing a decrease in social behavior in acute exposure while an increase in social behavior in chronic treatment. Moreover, they found that acute and chronic treatments differentially affected serotonergic neuronal populations in the dorsal raphe nucleus. In addition to the controversial outcomes, the effects of SSRIs on ASD are still debating, further studies are needed to clarify the association.

Serotonin (5-hydroxytryptamine, 5-HT), biosynthesized from the amino acid tryptophan, is mainly found in serotonergic neurons, enterochromaffin cells, and blood platelets. 5-HT is a crucial hormone and neurotransmitter that controls a number of neurobiological processes in the central nervous system (CNS). These processes are mediated by a group of serotonin receptors (5-HTRs), which are classified into seven subfamilies (5-HT_1–7_R). The 5-HTRs belong to a family of G protein-coupled receptors (GPCRs), except for 5-HT_3_R, which is a ligand-gated ion channel. 5-HT system plays a fundamental role in neurite outgrowth, dendritic spine morphology, shaping neuronal circuits, synaptic transmission, and synaptic plasticity. Given the roles of 5-HT, a variety of diseases such as depression, anxiety, schizophrenia, and neurodevelopmental disorders (e.g., ASD, fragile X syndrome, and Rett syndrome) have a close relation with regulation of 5-HT systems [33,34,35,36]. As therapeutic targets in numerous brain diseases, 5-HT has thus become a highly important class of biogenic amines. In this review, we will focus on recent studies on 5-HTR modulators as ASD therapeutics.

## 2. 5-HT_1_ Receptor (5-HT_1_R)

5-HT_1_Rs consist of the largest class of 5-HTR subtypes, namely, five subtypes such as 5-HT_1A_R, 5-HT_1B_R, 5-HT_1D_R, 5-HT_1E_R, and 5-HT_1F_R and distribute in the CNS including brain. Similar to other 5-HTRs, all of 5-HT_1_Rs show a nanomolar affinity for 5-HT. They couple to G_αi_/G_α0_ proteins to inhibit adenylyl cyclase and modulate subsequent signaling pathway and ionic effectors such as potassium or calcium channel [37]. Since they were discovered, numerous investigations with selective agonists and antagonists for 5-HT_1_Rs have been pursued and clear-cut evidence of the implication of them in anxiety behaviors, cognitive performance and neurobehavioral abnormality in animal study suggests that they could be potential therapeutic targets in psychopathological and neurodevelopmental diseases.

Khatri and co-workers studied the lasting effects of neonatal activation of 5-HT_1A_R and 5-HT_1B_R in a rat model [38]. Using day 8 to 21 postnatal rat pups, the authors investigated both agonist and antagonist effects for 5-HT_1A_R and 5-HT_1B_R with the administration of various ligands such as 8-hydroxy-2-(di-n-propylamino)-tetraline (8-OH DPAT, 5-HT_1A_R agonist) (Table 1), or CGS-12066B (5-HT_1B_R agonist)(Table 1), WAY-100635 (5-HT_1A_R antagonist) (Table 1), or GR-127935 (5-HT_1B_R antagonist) (Table 1) or citalopram, a well-known SSRI. Similar to the treatment of SSRI, the agonists show increased stereotypic activity and impaired social interactions similar to that observed in ASD. On the contrary, the selective antagonists alleviate the abnormal behaviors in the presence of SSRI. Their direct and indirect findings suggest that both receptors may have important roles in brain development even after birth.

Hollander and colleagues investigated the relationship between repetitive behavior and growth hormone with sumatriptan (Table 1), a known migraine treatment and a 5-HT_1D_R agonist [39]. The authors reported that administration of sumatriptan increases growth hormone level in blood, which ameliorates repetitive behavior in adult autistic patient. Though the research was conducted with only small number of men and failed to link the 5-HT level in blood, the discovery that 5-HTRs could be a therapeutic target for ASD is clinically significant.

De Boer and co-workers pursued behavioral studies with F15599 (Table 1) and F13714 (Table 1), 5-HT_1A_R agonists, specifically G_αi_ biased compounds, in male wild-type Groningen rats [40]. In addition to mitigating aggressive behavior, both agonists showed reduced social interaction and concomitant motor inactivity in a dose dependent manner. However, the authors claimed that the effect of agonists was fully abolished when WAY-100635, a 5-HT_1A_R antagonist, was pretreated. The results imply that 5-HT_1A_R-G_αi_ signaling pathway is especially mediated and that could be a worthy target for study.

Cosi and colleagues investigated profile of F17464 (Table 1), a partial 5-HT_1A_R agonist and human dopamine receptor subtype 3 agonist [41]. In addition to exhibiting good binding affinity (*K*_i_ = 0.16 nM for 5-HT_1A_R) and comparable activity to well-known drugs such as aripiprazole (Table 1), lurasidone (Table 1), and etc., lack of high binding affinity toward 5-HT_2A_R has attracted much attention as an antipsychotic drug candidate without side effects. The authors further reported that F17464 rescued impaired social interaction in valproate-treated rats, an animal model of autism [42]. Their findings suggest that 5-HT_1A_R plays an important role in social behavior.

An interesting study addressing the mechanism of action of psychostimulants such as amphetamine and cocaine on suppressed social interaction was reported by Achterberg and co-workers [43]. A thorough combinational study with various receptor antagonists, the authors revealed that amphetamine depends on alpha-2 noradrenergic receptor while cocaine involves in dopamine, noradrenaline, and 5-HTRs. Though further studies should be followed, the negative effects of psychostimulant drugs and their involvement in the serotonergic system may provide the glimpse of the etiology of ASD. 

Mogha and colleagues studied the effects of 5-HT_1A_R on signaling cascade using the C57BL/6 mice model [44]. The authors found that 5-HT_1A_R agonists increased PSD95 expression, a postsynaptic marker of spine and synapse maturation, and boosted dendritic spine and synapse formation via the sequential activation of the mitogen-activated protein kinase isozymes Erk1/2 and protein kinase C (PKC) in hippocampal region. Moreover, they observed that the stimulation of PKC restored the PSD95 expression and synaptogenesis in 5-HT_1A_R knockout mice. Their findings clearly show the importance of 5-HT_1A_R in signaling cascade to regulate hippocampal sculpting and function and indicate that 5-HT_1A_R has a crucial role in brain development. 

Lawson and co-workers examined the effect of 5-HT_1B_R agonist, RU24969 (Table 1), on the behavioral patterns in C57BL/6J mice [45]. The mice treated with RU24969 show reduced sociability and preference for social novelty in three chamber test. Moreover, the agonist decreases average rearing duration, which is a putative measurement of non-selective attention in mice. Deficit of the behavior is thought to be related to ASD. Taken together, the results indicate that the activation of 5-HT_1B_R induces ASD-like behavior in mice. Interestingly, the authors rescued the ASD-like behavior in mice with oxytocin, a hypothalamic neuropeptide. The findings suggest that oxytocin might restore the social deficits through a mechanistic pathway involving 5-HT_1B_R and that targeting 5-HT_1B_R is a worthwhile strategy to treat ASD. 

King and colleagues researched that fetal exposure to SSRI not only increases the serotonin level but also modulates p11 expression and neurogenesis in mice [46]. p11 protein was recently considered a crucial component of 5-HT signaling pathway through the complex formation with 5-HT_1B/1D_ receptors in adult [47]. The authors observed that p11 protein was detected in most of the fetal brain including thalamus, cortex, and hindbrain. Interestingly, they showed SSRI exposure in utero reduced p11 expression only in thalamus and decreased neurogenesis in hippocampal NE and galionic eminences, indicating that maternally administered SSRI could reach the fetal brain and modify p11 expression and neurogenesis. Although they failed to demonstrate the direct linking between p11 and 5-HT_1B/1D_ receptors, their results suggest that both receptors might influence p11 expression, which is important for neuronal development.

Rather than rodents, primate could be applied as a model animal. Larke and co-workers sought to evaluate social behavior in pair-bonded male titi monkeys when 5-HT_1A_R activity was modified by 8-OH-DPAT (Table 1), a selective agonist [48]. Contrary to many results in rodents, they monitored and quantified that the agonist led to at least two-fold decrease in affiliative social behaviors directed toward the pair-mate. Moreover, they found that the agonist reduced plasma concentration of cortisol without altering peripheral oxytocin level, which is increased in rodent cases. In addition to studying the effects of 5-HT_1A_R on behavior, their results are interesting not only using a monogamous primate for the first time, but also indicating a 5-HT_1A_R agonist affects social behavior differently between rodents and primates. 

Wu and colleagues investigated the effect of deep brain stimulation (DBS) method and the mechanism in autistic animal model [49]. DBS is an emerging therapeutic approach for Parkinson’s disease, depression, and other psychiatric diseases. By applying DBS method in infralimbic prefrontal cortex (ILPFC) of a valproate (VPA)-induced rat, the authors observed reduced behavioral abnormalities. The autistic behaviors were further improved with the combination of DBS and 8-OH-DPAT (Table 1), a 5-HT_1A_R agonist. Moreover, they reported that the expression level of glutamatergic and GABAergic neurons in PFC region was restored. They further confirmed that DBS treatment was unsuccessful in the presence of WAY100635 (Table 1), a 5-HT_1A_R antagonist. All of their results suggest that DBS treatment is effective via the 5-HT_1A_R system. 

Witt and co-workers tried to validate the effectiveness of vortioxetine (Table 1), a multimodal antidepressant, acting as an agonist for 5-HT_1A_R and 5-HT_1B_R and an inhibitor for 5-HT transporter [50]. The authors used BTBR T^+^Itpr3^tt^/J mice that lack social interaction preference and exhibit restrictive-repetitive behaviors. They observed high dose of vortioxetine could effectively decrease marble burying behaviors, and enhance social interaction although the effect was transient. Two-fold difference of vortioxetine preference toward 5-HT_1A_R and 5-HT_1B_R and the observation of low serum oxytocin level might be the reason of transiently enhanced sociability. Their results indicate that small molecules, which balance the nature to 5-HT_1A_R and 5-HT_1B_R, can be potential candidates for ASD by suppressing restrictive-repetitive behaviors and rescuing sociability.

SERT gene variant study by Veenstra-VanderWeele and colleagues suggested that 5-HT levels were closely associated with ASD [24]. The authors used SERT Ala56 variant from the native mouse *Slc6a4* locus, which exhibit normal growth and fertility but show enhanced CNS 5-HT clearance, increased 5-HTR sensitivity, and hyperserotonemia. Significantly, enhanced phosphorylation of p38 MAPK-dependent transporter and CNS 5-HT clearance in SERT Ala56 model appear to decrease synaptic 5-HT availability and thus increase a compensatory 5-HTR sensitivity. They further validated the effect with small molecules, 1-(2,5-dimethoxy-4-iodophenyl)-2-aminopropane (DOI) and 8-OH-DPAT, 5-HT_2A/2C_ and 5-HT_1A/7_ receptor agonist, respectively. Moreover, in their model, the authors observed a decrease in social interaction and an increase in repetitive behaviors, a stereotypy of ASD.

**Table 1 ijms-23-06515-t001:** 5-HT_1_R-targeted pharmacological agents which have potential effects on ASD treatment.

Names	Structures	Targets	Effects
8-OH-DPAT [24,38,44,48,49]	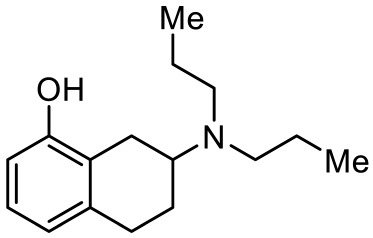	5-HT_1A_R/5-HT_7_R agonist	Increased stereotypic behavior and impaired social interaction via the disruption of sensory processing; enhanced dendritic spine and synapse formation; rescued sociability deficits, anxiety and hyperactivity with DBS treatment
CGS-12066B [38]	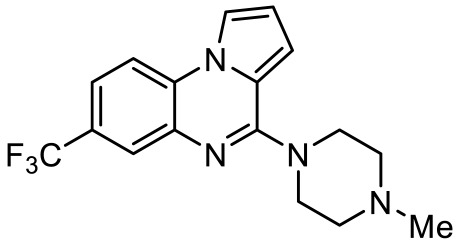	5-HT_1B_R agonist	Increased stereotypic behavior and impaired social interaction via the disruption of sensory processing
WAY-100635 [38]	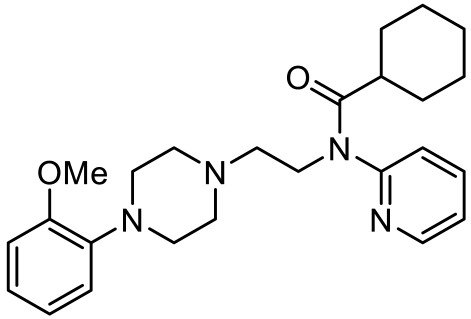	5-HT_1A_R antagonist	Slightly alleviate stereotypic behavior in the presence of SSRI
GR-127935 [38]	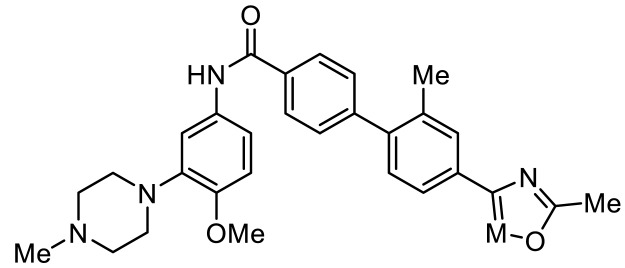	5-HT_1B_R antagonist	Slightly alleviate stereotypic behavior in the presence of SSRI
Sumatriptan [39]	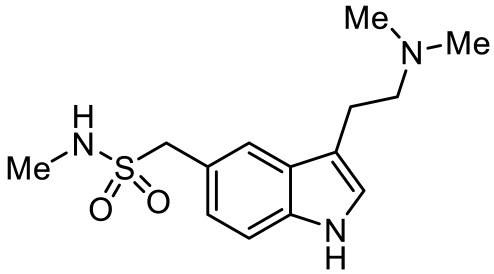	5-HT_1D_R agonist	Reduce repetitive behavior via an increase of growth hormone
F15599 [40]	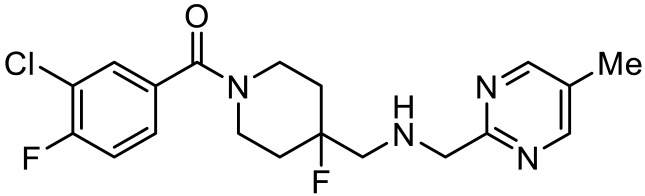	5-HT_1A_R agonist	Anti-aggressive effects and reduction of social interaction and an increase of motor inactivity
F13714 [40]	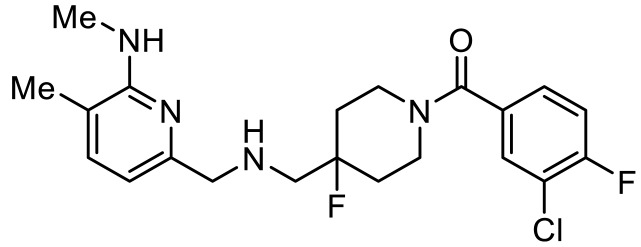	5-HT_1A_R agonist	Anti-aggressive effects and reduction of social interaction and an increase of motor inactivity
F17464 [41]	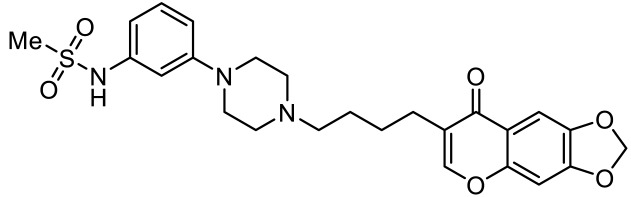	5-HT_1A_R partial agonist/D_3_R antagonist	Increases dopamine release, rescues impaired social interaction
Aripiprazole [10,11,12,13,14]	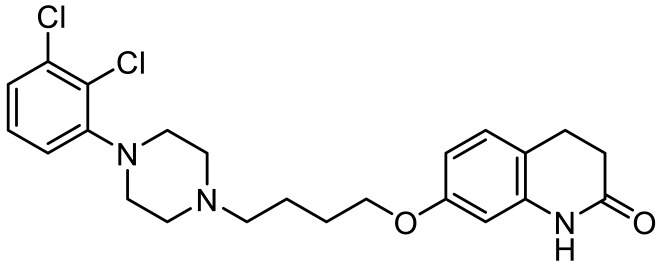	Affinity to 5-HT_1A_R/5-HT_2A_R/5-HT_2B_R/5-HT_7_R; D_2_R/D_3_R/D_4_R	Irritability amelioration
Lurasidone [10,11,12,13,14]	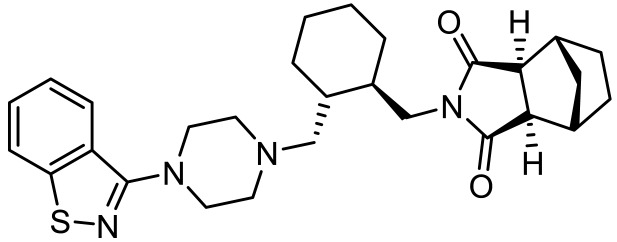	5-HT_2A_R/5-HT_7_/D_2_R antagonist; 5-HT_1A_R partial agonist	Irritability and aggressive behavior amelioration
RU24969 [45]	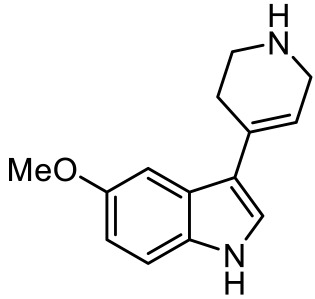	5-HT_1B_R agonist	Induce autism-like behavior including a decrease in social interaction
Vortioxetine [50]	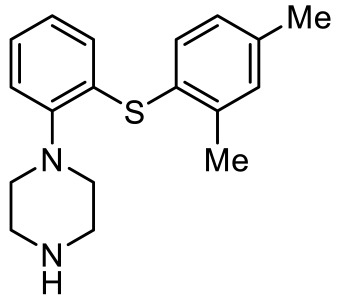	5-HT_1A_R/5-HT_1B_R agonist/inhibitor for SERT	Transiently suppress repetitive behavior and enhance social interaction

## 3. 5-HT_2_ Receptor (5-HT_2_R)

5-HT_2_ receptor is a member of the rhodopsin family of GPCRs, which are coupled with G_q_/G_11_, and consists of three subtypes, 5-HT_2A_R, 5-HT_2B_R, and 5-HT_2C_R. 5-HT_2A_R and 5-HT_2C_R are widely distributed in the CNS, while 5-HT_2B_R shows a restricted expression pattern in the CNS. 5-HT_2_Rs have been known to regulate a variety of signaling pathways in neurons including phospholipase C (PLC), phospholipase A2 (PLA2), and extracellular signal-regulated kinases (ERK) pathway [51,52]. The regulation of 5-HT_2_Rs is involved in the neuronal functions and activities, which are related with social behavior, cognitive function, anxiety, and the other neurodevelopmental disorders.

Molecular imaging studies of serotonin in ASD become crucial evidence of serotonergic abnormalities in the CNS of autistic individuals [53]. Initially, Chugani et al. used α-[^11^C]-methyl-L-tryptophan as a positron emission tomography (PET) tracer for serotonin synthesis [54]. This study discovered asymmetric serotonin synthesis in frontal cortex, thalamus, and dentate nucleus of the cerebellum in autistic children. In most cases, decreased serotonin synthesis was observed in frontal cortex and thalamus, whereas increased serotonin synthesis was observed in dentate nucleus. Moreover, the same research group found that the enhancement of capacity of brain serotonin synthesis during childhood is disrupted in autistic children [55]. In particular, Goldberg et al. performed a PET study using [^18^F]setoperone to image 5-HT_2_Rs in parents of autistic children [56]. This study demonstrated that cortical 5-HT_2_ binding potentials were notably lower in the parents compared with controls. Beversdorf et al. also used [^18^F]setoperone as the PET tracer to visualize 5-HT_2_Rs in adults with ASD [57]. The authors observed that thalamic [^18^F]setoperone binds less to 5-HT_2_Rs in autistic individuals than controls.

In addition, recent studies applying polymorphisms in the serotonin 2A receptor gene (*HTR2A*) in patients proved that 5-HT_2_R is genetically associated with ASD. Gadow et al. let parents of autistic children rate depressive symptoms of their children and then investigated association between the scores and functional single nucleotide polymorphisms in the *HTR2A*s, rs6311 and rs6314. The authors found that rs6311 may control the severity of depression in autistic children [58]. Smith et al. also supported that the rs6311 is associated with ASD based on the mRNA expression analysis and family-based clinical studies with ASD patients [59]. Given the molecular imaging and genetic evidence between serotonergic systems and ASD, 5-HT_2_Rs have been considered as important targets for treating ASD.

Risperidone and aripiprazole are only approved by the FDA for use in ASD to date. These drugs have been classified as atypical antipsychotics and used to treat irritability, self-injury, as well as aggression but not the core symptoms of ASD. Other atypical antipsychotics such as olanzapine, paliperidone, and cariprazine have been used or tested for treatment in ASD. Atypical antipsychotics generally show antagonist or partial agonist activity on both dopamine- and serotonin-receptors including 5-HT_2_Rs. Olanzapine (Table 2) acts as an antagonist at 5-HT_2A/2C_ and D_2_ dopamine receptors. When olanzapine was treated in autistic individuals, significant improvements in stereotyped behavior, social deficits, hyperactivity, irritability and aggression were observed in several case studies. However, it is known to cause various adverse effects [60]. Stigler et al. performed open-label study of paliperidone (9-hydroxyrisperidone) (Table 2), a 5-HT_2A_R antagonist, in 25 autistic individuals. Among the 25 subjects, 84% showed improvement based on a Clinical Global Impressions-Improvement (CGI-I) Scale. This study suggested that paliperidone treatment is significantly efficacious for irritability in autistic patients [61]. Cariprazine (Table 2), approved for treatment of schizophrenia and bipolar disorder, is a D_3_R, D_2_R, and 5-HT_1A_R partial agonist. The compound also acts as an antagonist at 5-HT_2A_R and 5-HT_2B_R. Román et al. very recently evaluated the cariprazine as treatment in autistic-like rats induced by the prenatal VPA exposure. The authors found that cariprazine is more efficient in the social play than risperidone and aripiprazole. In addition, cariprazine showed similar efficacy on behavioral tests to risperidone and aripiprazole (Table 1) [62]. Since atypical antipsychotics act at multiple receptor sites including dopamine- and serotonin-receptors as mentioned earlier, the drugs are able to cause various side effects such as weight gain, fatigue, gastrointestinal symptoms, hyperprolactinemia, sedation, and extrapyramidal symptoms. Taken together, more targeted approach using selective ligands is highly required to develop treatment minimizing adverse effects in ASD.

The selective 5-HT_2A_R antagonist M100907 (Table 2) has been applied to a potential therapeutic for ASD. Amodeo et al. investigated effects of risperidone and M100907 on the probabilistic reversal learning in autism mouse models, BTBR T^+^tf/J and C57BL/6J (B6). Both of the drugs enhanced reversal learning in BTBR mice, whereas higher dose of risperidone in B6 mice impaired reversal learning [63]. Treatment of M100907 was also efficient for attenuating repetitive grooming behavior in the BTBR mice, although meaningful effect on locomotor activity was not observed [64]. The same research group demonstrated that microinfusion of M100907 into the dorsomedial striatum in the BTBR mouse strain reduced reversal learning impairment and grooming behavior [65]. Panzini et al. studied the effect of M100907 in 16p11.2 deletion model mouse (Del mouse). Copy number variations of the region 16p11.2 are associated with ~1% of autism-related disorders. M100907 returned to normal level of active coping and improved the gradual shift to passive coping in Del mice [66].

In addition, Baker et al. investigated effects of another 5-HT_2A_R antagonist, ketanserin (Table 2), on strategy-switching in Long-Evans rats. Administration of ketanserin improved behavioral switching between visual cue and response strategy [67]. Given the beneficial effects of 5-HT_2A_R antagonists in various autism mouse models, the selective 5-HT_2A_R antagonists become promising potential therapeutics for ASD.

One research investigated how activation of 5-HT_2A_R and 5-HT_2C_R affects behavioral flexibility in an autism mouse model, B6 mouse. DOI (a 5-HT_2A/2C_ receptor agonist)(Table 2), 25CN-NBOH (a selective 5-HT_2A_R agonist) (Table 2), and SER-082 (a 5-HT_2C_R antagonist)(Table 2) were used to demonstrate that 5-HT_2A_R agonism impairs probabilistic reversal learning. Treatment of B6 male mice with DOI alone did not impair behavioral flexibility, while cases of combination between SER-082 and DOI showed the impairment. Moreover, administration of 25CN-NBOH elevated the number of trials to criterion on a reversal learning task. This study thus proposed that 5-HT_2A_R activation causes impairments on probabilistic reversal learning while co-activation of 5-HT_2C_R is probably able to prevent the impairments [68]. Previous studies on association between 5-HT_2A_R and 5-HT_2C_R supported that these two receptors bring opposing effects on behavioral flexibility [69,70].

Some reports showed the contrary results to what 5-HT_2A_R activation produces on social behavior in autism mouse models. Knöpfel et al. investigated effects of psilocybin (Table 2), which is known as a serotonergic psychedelic drug and a 5-HT_2A_R agonist, on social behavior in B6 mouse model exposed prenatally to VPA. To exclude acute psychedelic effects, the authors performed sociability and social memory behavioral tests after 24 h following administration of psilocybin. The results showed that psilocybin ameliorate the social behavioral abnormalities in the autism model mice [71]. In addition, treatment of 5-HT_2C_R antagonist SB242084 (Table 2) in the ASD risk gene *Pten* haploinsufficient mice rescued the social behavior deficits [72], whereas one research reported that SB242084 did not enhance strategy switching [67].

**Table 2 ijms-23-06515-t002:** 5-HT_2_ receptors-targeted pharmacological agents which have potential effects on ASD treatment.

Names	Structures	Targets	Effects
Olanzapine [60]	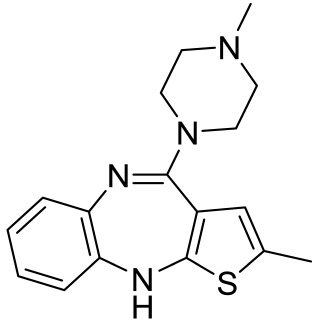	5-HT_2A_R/5-HT_2C_R/D_2_R antagonist	Irritability amelioration
Paliperidone [61]	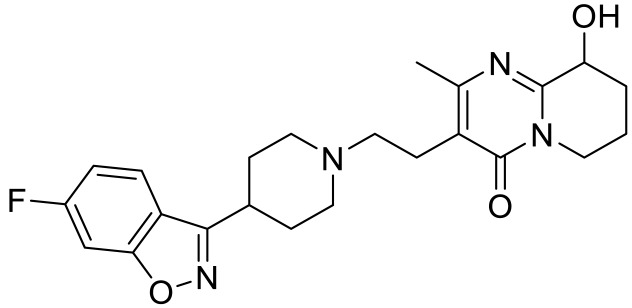	5-HT_2A_R antagonist	Significant improvement in irritability
Cariprazine [62]	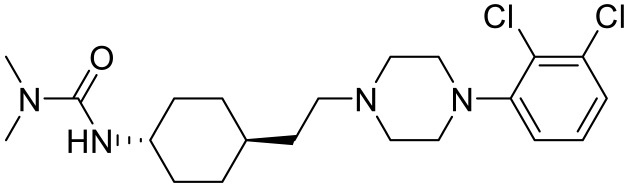	5-HT_2A_R/5-HT_2B_R antagonist, 5-HT_1A_R/D_2_R/D_3_R partial agonist	Alleviation of ASD core behavioral deficits
M100907 [63,64,65,66]	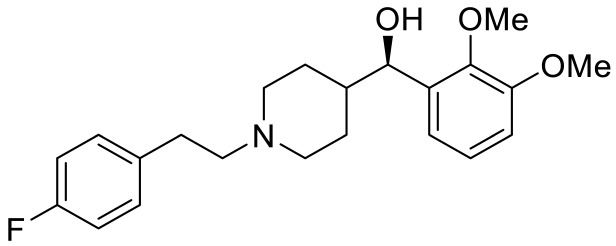	5-HT_2A_R antagonist	Improvement of probabilistic reversal learning; reduced repetitive behaviors
Ketanserin [67]	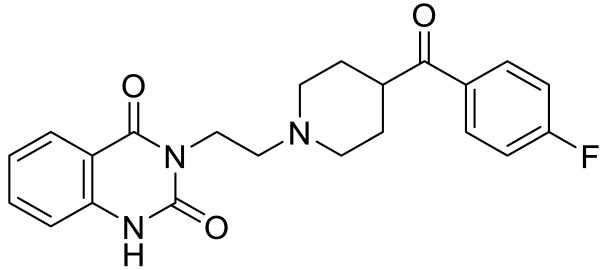	5-HT_2A_R/H_1_R antagonist	Enhanced strategy-switching between a visual cue and response strategy
25CN-NBOH [68]	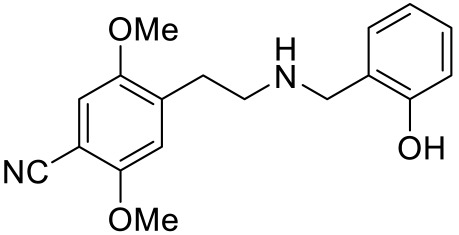	5-HT_2A_R agonist	Impairment of probabilistic reversal learning
DOI and SER-082 [68]	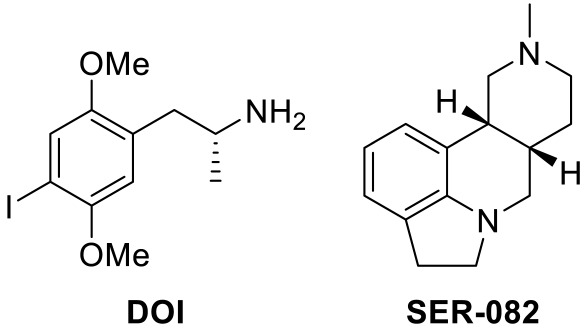	DOI: 5-HT_2A_R/5-HT_2C_R agonist; SER-082: 5-HT_2B_R/5-HT_2C_R antagonist	DOI alone did not impair reversal learning; co-treatment of DOI with SER-082 impair reversal learning
Psilocybin [71]	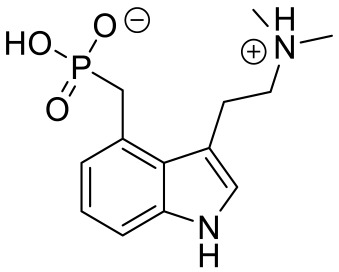	5-HT_2A_R agonist	Rescued the social behavioral abnormalities
SB242084 [67,72]	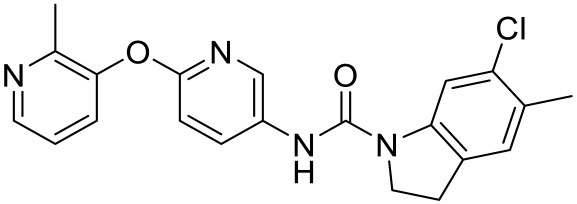	5-HT_2C_R antagonist	Rescued social behavior deficits

## 4. 5-HT_6_ Receptor (5-HT_6_R)

5-HT_6_R is one of the latest discovered serotonin receptors and mainly localized in the striatum, cerebral cortex, as well as hippocampus [73,74,75,76]. These brain regions play a critical role in working memory, learning, and cognitive flexibility related with neurodevelopmental disorders including ASD [77,78]. Gs-adenyl cyclase activity is stimulated by 5-HT_6_R leading to cyclic adenosine monophosphate (cAMP) release [79]. 5-HT_6_ receptor also interacts with the tyrosine kinase Fyn that regulates cell proliferation, survival, and death [80]. Moreover, a variety of signaling pathways such as the ERK1/2 pathway, the Jun activation domain-binding protein 1 (Jab1) pathway, the microtubule-associated protein (Map1b) pathway, and the mTOR pathway are involved in 5-HT_6_R signaling [81,82,83]. In particular, the mTOR pathway is known to be closely related with neurodevelopmental mechanisms [83]. Furthermore, 5-HT_6_R signaling pathways regulate other neurotransmitter systems such as cholinergic, glutamatergic, and dopaminergic systems, which may be linked with cognitive effects and mood regulation [84,85,86]. Given the roles of 5-HT_6_R in neurotransmitter systems, 5-HT_6_R has recently emerged as an interesting target for various brain disorders [87,88].

Ragozzino et al. demonstrated that PRX-07034 (Table 3), having an aryl piperazine core, is a highly potent and selective 5-HT_6_R antagonist compared to other GPCRs, ion channels, and transporters including other 5-HTRs [89]. PRX-07034 alleviated cognitive flexibility impairments and enhanced working memory as well as strategy switching in male Long-Evans rats. 5-HT_6_R blockade may affect acetylcholine release in the frontal cortex and striatum or directly glutamate transmission. The results thus suggested that PRX-07034 is probably effective for individuals having working memory impairments such as in ASD and schizophrenia.

Administration of other aryl piperazine 5-HT_6_R antagonists showed similar effects with PRX-07034 such as enhanced cognitive functions and memory consolidation. Bécamel et al. found that treatment of adult mice, which are exposed to cannabis during adolescence, with SB-258585 (Table 3) plays a crucial role to control synaptic plasticity, excitatory/inhibitory balance, and intrinsic neuronal properties via 5-HT_6_R-operated mTOR signaling network. The authors suggested that 5-HT_6_R antagonists might be evaluated in ASD based on the evidence of close relationship between mTOR and ASD [90]. In addition, treatment of SB-271046 (Table 3) and SB-399885 (Table 3) reversed scopolamine-induced deficits in aged rats. These compounds also improved task acquisition and ameliorated spatial task deficits [91,92]. 

Amodeo et al. reported that 5-HT_6_R blockade attenuated core symptoms of ASD in very recent [93]. To prove the assumption, BTBR T^+^Itpr3tf/J and C58/J (C58) mouse strains, showing repetitive behaviors and social deficits, had been used. Treatment of BTBR mice with a selective 5-HT_6_R antagonist BGC20-761 (Table 3) attenuated repetitive grooming activity regardless of sex while treatment of C58 mice with BGC20-761 increased social sniff time in female. The authors explained that 5-HT_6_R blockade promotes acetylcholine release leading to attenuation in repetitive behaviors in BTBR mice. Moreover, Neumaier et al. supported that BGC20-761 enhanced memory consolidation and cognitive function in rodents having social recognition deficits induced by an acetylcholine antagonist scopolamine, although BGC20-761 was not effective to attenuate anxiety behaviors [94].

Interestingly, one research investigated that activation of 5-HT_6_R probably impairs working memory and behavioral flexibility. When a 5-HT_6_R agonist EMD386088 (Table 3) was administrated in B6 mice, impairments of probabilistic reversal learning performance and spontaneous alternation performance were observed [95]. Given the recent findings of selective 5-HT_6_R antagonists on ASD core symptoms, the development of highly selective 5-HT_6_R modulators is desired for ASD therapeutics.

## 5. 5-HT_7_ Receptor (5-HT_7_R)

5-HT_7_ receptor is the most recently discovered 5-HT receptor among 5-HTR subtypes, which is largely distributed in the various area of brain such as thalamus, hypothalamus, and cortex, and regulates brain development, synaptic transmission and plasticity, learning and memory [96]. Four isoforms of 5-HT_7_R have been reported, namely, 5-HT_7a_R, 5-HT_7b_R, 5-HT_7c_R, and 5-HT_7d_R, and among them, only 5-HT_7c_R is not detected in human [97]. Similar to other 5-HTR, 5-HT_7_R is coupled to G_s_ protein, resulting in an increase of cAMP via the activation of adenylyl cyclase (AC). It can also recruit G_12_ protein, one of G_α_ subunits, which interacts with various RhoGEFs, inducing the activation of Rho GTPases [98]. Although a single base polymorphism for 5-HT_7_R was detected, an analysis of transmission disequilibrium in autistic patients failed to provide any correlation of allele to autism [99]. However, even though the dysfunction of 5-HT_7_R has not been reported in ASD patients, the distribution of 5-HT_7_R in functional brain area and some evidence implicate their involvement, as discussed below.

Khodaverdi and co-workers reported that the administration of 5-HT_7_R agonist, LP211 (Table 4), for a week rescues all behavioral deficits in an autistic-like rat model induced by an anti-migraine medication, VPA [100]. The authors revealed that the 5-HT_7_R agonist also restored the impaired synaptic plasticity in hippocampal region. Their findings clearly suggest that the serotonergic system, 5-HT_7_R, could be a promising pharmacotherapy target for ASD.

Canal and colleagues studied an aminotetralin derivative, (+)-5-FPT (Table 4), which is motivated from AS-19 and 8-OH-DPAT (Table 1) [101]. The compound has high affinity for 5-HT_7_R and 5-HT_1A_R and works as a partial agonist. In vivo studies in three heterogeneous mouse models revealed that the compound not only reduced stereotypy of ASD but also increased social interaction without causing significant side effects. Moreover, the authors pursued thorough pharmacokinetic study and claimed that orally operative (+)-5-FPT appears to be a promising candidate for ASD treatment. The group re-evaluated pharmacokinetic values and further validated the effect of (+)-5-FPT in a genetic model, *Fmr1* knockout, and wild-type mice [102]. In addition to providing anxiolytic-like effects, it significantly relieved repetitive behavior and enhanced social interaction both in wild-type and the transgenic mice. After developing the lead compound, (+)-5-FPT, [101] Perry et al. synthesized more derivatives and pursued 3D-QSAR and molecular docking studies to understand important structural requirements. The authors found that a steric appendage either at C5- or chiral C2-position enhances selectivity for 5-HT_7_R or 5-HT_1A_R, respectively [103].

Many groups have tried to develop small molecules with fancy core structure. Motivated from bioactive molecules, Kelemen et al. synthesized and performed preliminary SAR (structure-activity relationship) analysis with spiro-oxindole derivatives [104]. Many of the compounds exhibit high affinity and selectivity for 5-HT_7_R over other 5-HTR subtypes. By exploiting knowledge-based design strategy, Lacivita and colleagues synthesized a variety of long-chain arylpiperazine derivatives and displayed interesting activity profiles such as dual 5-HT_7_/5-HT_1A_ receptor agonist properties and mixed 5-HT_7_/5-HT_1A_ receptor agonist/5-HT_2A_R antagonist properties [105]. The authors claimed that the compounds are metabolically stable in vitro and especially exhibit druglike properties, therefore, they could be potential candidates for ASD treatments.

As many studies on the GPCR signaling pathway have progressed, it has been found that not only G proteins but non-G protein factors such as *β*-arrestin are signal inducers in the signaling pathway [106]. To understand the mechanism well, ligands with functional selectivity or signaling biased ligands have been actively developed. Our group also has pursued biased ligands for 5-HT_7_R to understand its mechanism in ASD. Kim et al. developed a series of tetrahydroazepine derivatives with arylpyrazolo moiety or arylisoxazolo moiety consideration of pyrazole-based molecules [107] such as agonists E55888 [108] and AS-19 [109] and antagonist JNJ18038638 [110]. Among them, 3-(4-chlorophenyl)-1,4,5,6,7,8-hexahydropyrazolo [3,4-*d*] azepane acts as the selective and *β*-arrestin biased ligand for 5-HT_7_R. Although thorough studies were not pursued, modeling comparison with AS-19 suggested that ionic interaction with Asp162 and π-alkyl interaction with Ile233 appear to be important for the *β*-arrestin signaling pathway. Lee et al. more focused on the biphenyl core structure with variety of amine scaffolds [111]. In contrast with E55888, a balanced agonist, 2-(6-chloro-2′-methoxy-[1,1′-biphenyl]3-yl)-*N*-ethylethan-1-amine (Table 4) shows G-protein biased activity without affecting *β*-arrestin signaling pathway. Interestingly, the compound increased the duration of self-grooming in Shank3 transgenic mice, a typical behavior in ASD. More recently, Kwag et al. discovered G-protein biased antagonists by modifying pyrazolyl-diazepanes and pyrazolyl-piperazines [112]. Among the compounds, 1-(3-(3-chlorophenyl)-1*H*-pyrazol-4-yl)-1,4-diazepane (Table 4) showed the best selectivity over any other 5-HTRs and binding affinity for 5-HT_7_R. The compound dramatically reduced self-grooming duration time in *Shank3* transgenic mice to the level of wild type mice. To understand why the compound acts as a G-protein biased antagonist, we further pursued docking analysis. While our previous compound forms an interaction with Ile233, [107] the compound resides in a hydrophobic pocket having ionic interaction with Asp162 and no interaction with Ile233, implying that Ile233 might be a crucial residue for *β*-arrestin signaling pathway. Both results of G-protein biased compounds suggest that 5-HT_7_R is associated with ASD and 5-HT_7_R could be a potential therapeutic target for the treatment of ASD.

**Table 4 ijms-23-06515-t004:** 5-HT_7_R-targeted pharmacological agents which have potential effects on ASD treatment.

Names	Structures	Targets	Effects
LP211 [100]	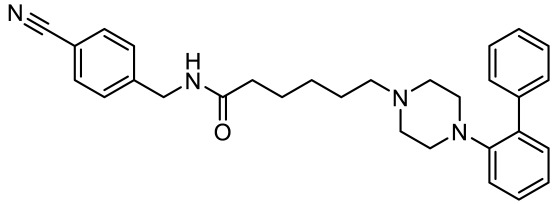	5-HT_7_R agonist	Rescue all behavioral deficits, and restores hippocampal synaptic plasticity impairment
(+)-5-FPT [101,102]	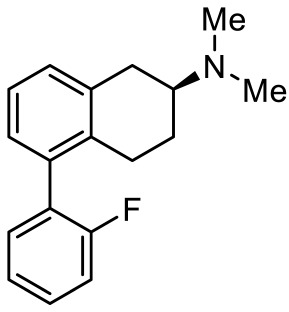	5-HT_1A_R/5-HT_2C_R/5-HT_7_R partial agonist	Reduction of stereotypic behavior, social activity increase; reduced the number of audiogenic seizures
2-(4-(4-(4’-methoxy-[1,1’-biphenyl]-2-yl)piperazin-1-yl)butyl)-4-methyl-1,2,4-triazine-3,5(2H,4H)-dione [105]	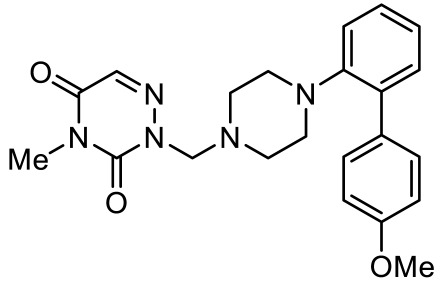	Dual 5-HT_1A_R/5-HT_7_R agonist	Metabolically stable and have suitable CNS druglike properties
6-((R)-2-hydroxy-3-(4-(4’-methoxy-[1,1’-biphenyl]-2-yl)piperazin-1-yl)propoxy)-2-methyl-2H-benzo[b][1,4]oxazin-3(4H)-one [105]	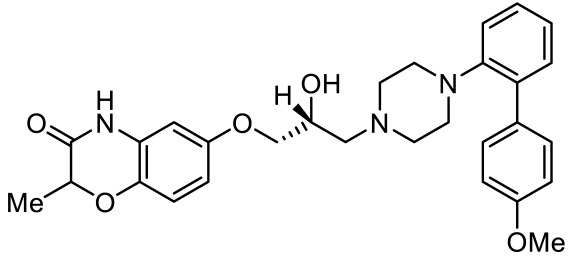	Dual 5-HT_1A_R/5-HT_7_R agonist	Metabolically stable and have suitable CNS druglike properties
7-(3-(4-(4’-methoxy-[1,1’-biphenyl]-2-yl)piperazin-1-yl)propoxy)-4-methyl-2H-chromen-2-one [105]	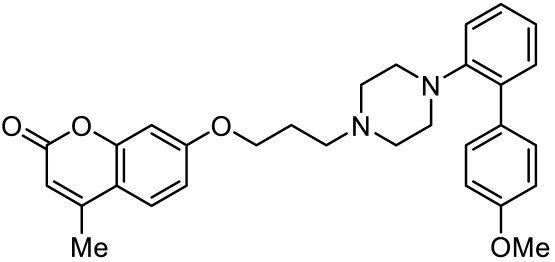	Mixed 5-HT_1A_R/5-HT_7_R agonist; 5-HT_2A_R antagonist	Metabolically stable and have suitable CNS druglike properties
3-(4-chlorophenyl)-1,4,5,6,7,8-hexahydropyrazolo[3,4-*d*] azepane [107]	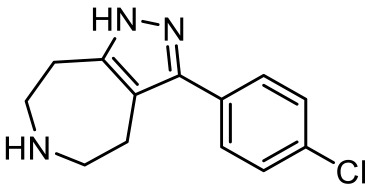	5-HT_7_R agonist (β-arrestin partial agonist)	Increase in non-rapid eye movement (NREM) sleep duration and decrease in REM sleep duration
2-(6-chloro-2′-methoxy-[1,1′-biphenyl]3-yl)-N-ethylethan-1-amine [111]	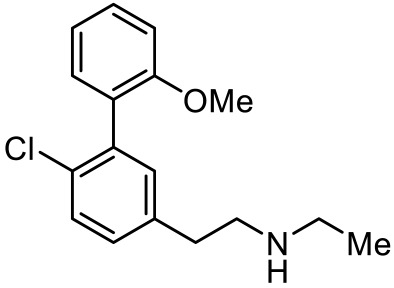	5-HT_7_R agonist (G protein biased agonist)	Increase in the duration of self-grooming
1-(3-(3-chlorophenyl)-1*H*-pyrazol-4-yl)-1,4-diazepane [112]	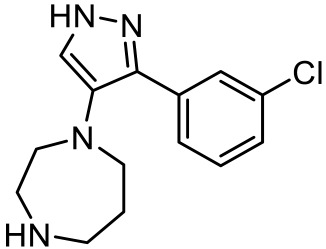	5-HT_7_R antagonist (G protein biased antagonist)	Decrease in the duration time of self-grooming

## 6. Conclusions and Future Perspectives

Among seven subfamilies of 5-HTRs, 5-HT_1_R, 5-HT_2_R, 5-HT_6_R, and 5-HT_7_R have been intensively investigated as potential targets for ASD. This review provides recent studies in development of small molecule modulators of these 5-HTRs as ASD therapeutic agents. A variety of selective agonists and antagonists for 5-HT_1_Rs have been developed and evaluated in autistic patients and some animal models to date. In particular, administration of 5-HT_1A_R, 5-HT_1B_R, and 5-HT_1D_R modulators ameliorate repetitive behavior and enhance social interaction which are core symptoms of ASD. PET studies of serotonin in ASD and polymorphisms in the *HTR2A* gene demonstrated that 5-HT_2_ receptors are associated with ASD. Afterward, numerous 5-HT_2_R modulators have been evaluated as ASD therapeutics. 5-HT_2A_R and 5-HT_2C_R antagonists including some atypical antipsychotics showed improvement in core symptoms in ASD patients and mice model, although one contrary case was reported that psychedelic prodrug psilocybin having 5-HT_2A_R agonism rescued social behavior deficits. Highly selective 5-HT_6_R antagonists such as PRX-07034, SB-258585, SB-271046, SB-399885, BGC20-761 enhanced cognitive function and memory as well as reduced repetitive behavior. Treatment of 5-HT_6_R agonist EMD386088 impaired behavioral flexibility and working memory in autism model mice. 5-HT_7_R modulators are known to improve altered behaviors and affect neuronal morphology. Recent development of G-protein biased ligands of 5-HT_7_R led to alleviation of self-grooming in autism mice. 5-HT_7_R thus becomes a potential therapeutic target for neurodevelopmental disorders including ASD [113].

Development of effective medications for ASD is challenging due to the properties of ASD such as the wide heterogeneity of factors and symptoms, absence of biomarkers, as well as different treatment outcomes depending on the age of patients. Despite continuous efforts for advancing ASD treatments, there are still only two FDA-approved drugs (risperidone and aripiprazole). Since the first classification of 5-HTR subtypes in 1957 [114], their structures, pharmacology, functions, signaling pathways, and clinical relevance have been disclosed [115]. As herein reviewed, 5-HTRs have recently emerged as potential targets for ASD because direct or indirect findings suggest that modulation of 5-HTRs is closely associated with ASD. In addition, as the targeted approach, treatments of selective modulators of 5-HTRs have been demonstrated to be effective for alleviating core symptoms of ASD and reducing side effects. Taken together, we believe that the development of a number of highly selective ligands for 5-HTRs will probably provide new opportunities for the discovery of ASD cures.

## Figures and Tables

**Table 3 ijms-23-06515-t003:** 5-HT_6_R-targeted pharmacological agents which have potential effects on ASD treatment.

Names	Structures	Targets	Effects
PRX-07034 [89]	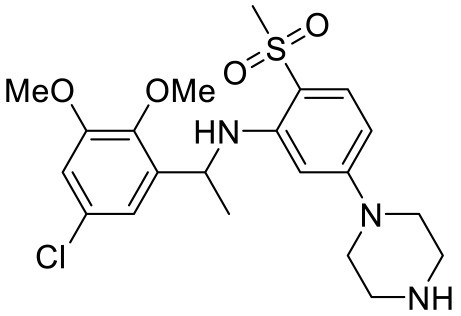	5-HT_6_R antagonist	Enhancing working memory and cognitive flexibility
SB-258585 [90]	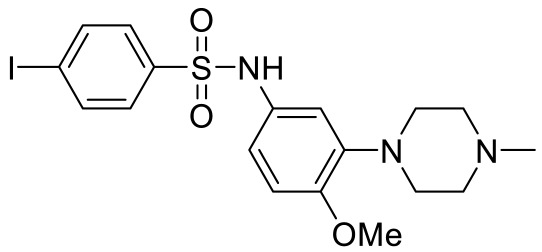	5-HT_6_R antagonist	Prevention of cognitive symptom onset
SB-271046 [91]	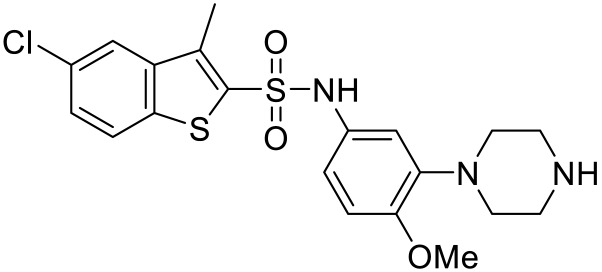	5-HT_6_R antagonist	Reversing the amnesia produced by scopolamine administration; amelioration of spatial task deficits
SB-399885 [92]	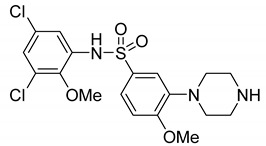	5-HT_6_R antagonist	Enhancing cognitive function
BGC20-761 [93,94]	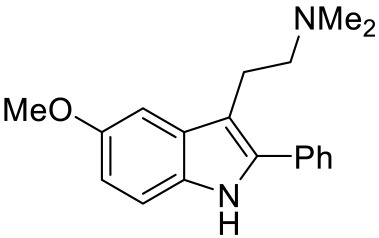	5-HT_6_R antagonist	Reduced repetitive behavior; enhancing memory consolidation, reversing scopolamine-induced memory deficit
EMD386088 [95]	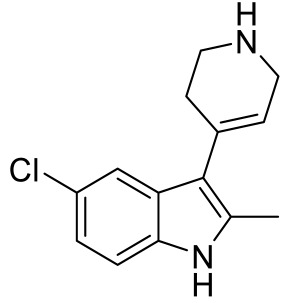	5-HT_6_R agonist	Impairments of behavioral flexibility and working memory

## Data Availability

Not applicable.

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
