# Peer review of "Serotonin Receptors as Therapeutic Targets for Autism Spectrum Disorder Treatment"

_ijms, 2022, doi:10.3390/ijms23126515_

Round 1

Reviewer 1 Report

The authors provide a current overview of the role of the serotonin system in ASD and potential therapeutics that target this neurotransmitter system. This review will serve as a suitable and comprehensive assessment of current knowledge.

One suggestion would be for the authors to provide a Future Directions section, which would help to clarify any gaps in our current understanding and possible next steps in regards to serotonin and ASD.

Author Response

Reviewer 1

Comment 1: The authors provide a current overview of the role of the serotonin system in ASD and potential therapeutics that target this neurotransmitter system. This review will serve as a suitable and comprehensive assessment of current knowledge.

Response 1: We appreciate this assessment.

Comment 2: One suggestion would be for the authors to provide a Future Directions section, which would help to clarify any gaps in our current understanding and possible next steps in regards to serotonin and ASD.

Response 2: We are grateful for this valuable comment. The conclusion was revised with an additional paragraph as follows on page 16. In addition, we changed the sixth heading from “6. Conclusion” to “6. Conclusions and Future Perspectives”.

“Development of effective medications for ASD is challenging due to the properties of ASD such as the wide heterogeneity of factors and symptoms, absence of biomarkers, as well as different treatment outcomes depending on the age of patients. Despite continuous efforts for advancing ASD treatments, there are still only two FDA-approved drugs (risperidone and aripiprazole). Since the first classification of 5-HTR subtypes in 1957, their structures, pharmacology, functions, signaling pathways, and clinical relevance have been disclosed. As herein reviewed, 5-HTRs have recently emerged as potential targets for ASD because direct or indirect findings suggest that modulation of 5-HTRs is closely associated with ASD. In addition, as the targeted approach, treatments of selective modulators of 5-HTRs have been demonstrated to be effective for alleviating core symptoms of ASD and reducing side effects. Taken together, we believe that the development of a number of highly selective ligands for 5-HTRs will probably provide new opportunities for the discovery of ASD cures.”

Reviewer 2 Report

Manuscript of   Ansoo Lee1, Hyunah Choo and Byungsun Jeon. “Serotonin Receptors as Therapeutic Targets for Autism Spectrum Disorder Treatment” is rather well written.

The review is easy to read, the authors managed to write it clear and quite interesting.

I would also advise to read the Barnes et al 2021 review, which describes all types of serotonin receptors very well and where they are located (on what cells, in what structures), since the authors write about the 6th receptor. That 5-HT6R signaling pathways regulate other neurotransmitter systems such as cholinergic, glutamatergic, and dopaminergic systems and about others receptors are not written such information.

Also authors should put in order the introduction of abbreviations along the text (I have indicated some below). Sometimes they have a broken order of appearance in the text, that is, the abbreviated name comes at the beginning, and the interpretation and the accepted abbreviation appear later.

There are some errors and typos in the text, and also extra spaces.  For example:

Line 35 – “There are only a couple of FDA-ap-35 proved drugs …” FDA -full name is required/ the first time in the text

Line 104 8-OH DPAT - full name is required/ the first time in the text

Line 207 -208 - 8-hydroxy-2-(di- 207 n-propylamino)-tetraline (8-OH-DPAT) –in this line autors can use only 8-OH DPAT

Line 245   Food and Drug Administration (FDA) –it is should be in line 35 –full name

Line 291 1-(2,5-Dimethoxy-4-iodophenyl)-2-aminopropane (DOI, a 5-HT2A/2C receptor agonist), DOI was explain in line 207 so this is unnecessary

Line 318, 323 -HT6 R change to HT6R

I think that the paper is o'key to be published after minor revision

Author Response

Reviewer 2

Comment 1: Manuscript of Ansoo Lee, Hyunah Choo and Byungsun Jeon. “Serotonin Receptors as Therapeutic Targets for Autism Spectrum Disorder Treatment” is rather well written.

The review is easy to read, the authors managed to write it clear and quite interesting.

Response 1: We appreciate this assessment.

Comment 2: I would also advise to read the Barnes et al 2021 review, which describes all types of serotonin receptors very well and where they are located (on what cells, in what structures), since the authors write about the 6th receptor. That 5-HT6R signaling pathways regulate other neurotransmitter systems such as cholinergic, glutamatergic, and dopaminergic systems and about others receptors are not written such information.

Response 2: We are grateful for this valuable comment and introduction of the nice review paper. To reflect this comment, we have added an additional sentence in the manuscript as follows on page 16 and we have included the references for the supportive information of serotonin receptors in the manuscript.

“Since the first classification of 5-HTR subtypes in 1957, their structures, pharmacology, functions, signaling pathways, and clinical relevance have been disclosed.”

Comment 3: Also authors should put in order the introduction of abbreviations along the text (I have indicated some below). Sometimes they have a broken order of appearance in the text, that is, the abbreviated name comes at the beginning, and the interpretation and the accepted abbreviation appear later.

There are some errors and typos in the text, and also extra spaces.  For example:

Line 35 – “There are only a couple of FDA-ap-35 proved drugs …” FDA -full name is required/ the first time in the text

Line 104 8-OH DPAT - full name is required/ the first time in the text

Line 207 -208 - 8-hydroxy-2-(di- 207 n-propylamino)-tetraline (8-OH-DPAT) –in this line autors can use only 8-OH DPAT

Line 245   Food and Drug Administration (FDA) –it is should be in line 35 –full name

Line 291 1-(2,5-Dimethoxy-4-iodophenyl)-2-aminopropane (DOI, a 5-HT2A/2C receptor agonist), DOI was explain in line 207 so this is unnecessary

Line 318, 323 -HT6 R change to HT6R

Response 3: Sorry for all these errors and typos, which have been fixed throughout the manuscript. In addition, we have added an additional content “Abbreviations” in the manuscript on page 16:

Abbreviations

ASD                                             autism spectrum disorder

FDA                                             Food and Drug Administration

5-HT                                            Serotonin, 5-hydroxytryptamine

5-HTR                                          5-HT receptor

SSRI                                             selective serotonin reuptake inhibitor

SERT                                            serotonin transporter

CNS                                             central nervous system

5-HTnR                                         5-HTn receptor

GPCR                                           G protein-coupled receptor

8-OH-DPAT                                8-hydroxy-2-(di-n-propylamino)-tetraline

PKC                                             protein kinase C

DBS                                              deep brain stimulation

ILPFC                                          infralimbic prefrontal cortex

VPA                                             valproate, valporic acid

DOI                                              1-(2,5-dimethoxy-4-iodophenyl)-2-aminopropane

PLC                                              phospholipase C

PLA2                                            phospholipase A2

ERK                                             extracellular signal-regulated kinases

PET                                              positron emission tomography

HTR2A                                        serotonin 2A receptor gene

CGI-I                                           Clinical Global Impressions-Improvement

cAMP                                           cyclic adenosine monophosphate

Jab1                                              Jun activation domain-binding protein 1

Map1b                                         microtubule-associated protein

AC                                               adenylyl cyclase

SAR                                              structure-activity relationship

Comment 4: I think that the paper is o'key to be published after minor revision

Response 4: We would like to express our appreciation for the supportive comment.